# The prone position in COVID-19 impacts the thickness of peripapillary retinal nerve fiber layers and macular ganglion cell layers

Niphon Chirapapaisan[1], Akarawit Eiamsamarng[1]*, Wanicha Chuenkongkaew[1‡], Natthapon Rattanathamsakul[2], Ranistha Ratanarat[3‡]

**1** Faculty of Medicine Siriraj Hospital, Department of Ophthalmology, Mahidol University, Bangkok, Thailand, **2** Faculty of Medicine Siriraj Hospital, Department of Medicine, Division of Neurology, Mahidol University, Bangkok, Thailand, **3** Faculty of Medicine Siriraj Hospital, Department of Medicine, Division of Critical Care, Mahidol University, Bangkok, Thailand

☾ These authors contributed equally to this work.
‡ These authors also contributed equally to this work.
* akarawit.eam@mahidol.ac.th

**Data Availability Statement:** All relevant data are within the paper and its supporting information files.

## Abstract

The prone position reduces mortality in severe cases of COVID-19 with acute respiratory distress syndrome. However, visual loss and changes to the peripapillary retinal nerve fiber layer (p-RNFL) and the macular ganglion cell layer and inner plexiform layer (m-GCIPL) have occurred in patients undergoing surgery in the prone position. Moreover, COVID-19-related eye problems have been reported. This study compared the p-RNFL and m-GCIPL thicknesses of COVID-19 patients who were placed in the prone position with patients who were not. This prospective longitudinal and case-control study investigated 15 COVID-19 patients placed in the prone position (the "Prone Group"), 23 COVID-19 patients not in the prone position (the "Non-Prone Group"), and 23 healthy, non-COVID individuals without ocular disease or systemic conditions (the "Control Group"). The p-RNFL and m-GCIPL thicknesses of the COVID-19 patients were measured at 1, 3, and 6 months and compared within and between groups. The result showed that the Prone and Non-Prone Groups had no significant differences in their p-RNFL thicknesses at the 3 follow-ups. However, the m-GCIPL analysis revealed significant differences in the inferior sector of the Non-Prone Group between months 1 and 3 (mean difference, 0.74 μm; $P = 0.009$). The p-RNFL analysis showed a significantly greater thickness at 6 months for the superior sector of the Non-Prone Group (131.61 ± 12.08 μm) than for the Prone Group (118.87 ± 18.21 μm; $P = 0.039$). The m-GCIPL analysis revealed that the inferior sector was significantly thinner in the Non-Prone Group than in the Control Group (at 1 month 80.57 ± 4.60 versus 83.87 ± 5.43 μm; $P = 0.031$ and at 6 months 80.48 ± 3.96 versus 83.87 ± 5.43 μm; $P = 0.044$). In conclusion, the prone position in COVID-19 patients can lead to early loss of p-RNFL thickness due to rising intraocular pressure, which is independent of the timing of prone positioning. Consequently, there is no increase in COVID-19 patients' morbidity burden.

**Funding:** This study was conducted with a grant from the Siriraj Research Fund, Faculty of Medicine, Siriraj Hospital, Mahidol University (IO–R016531014). The funders had no role in study design, data collection and analysis, decision to publish, or preparation of the manuscript.

**Competing interests:** The authors have declared that no competing interests exist.

## Introduction

A high mortality rate is associated with coronavirus disease 2019 (COVID-19) because the acute phase results in irreversible harm to multiple organs. Furthermore, the aftereffects of COVID-19 following recovery also result in increased morbidity.

In severe cases of COVID-19 involving acute respiratory distress syndrome, global data have established that the prone procedure lowers the mortality rate [1–3]. Many studies have also reported that perioperative visual loss has occurred in post-spinal operative patients whose surgery was performed in the prone position [4–6]. It has been suggested that ischemic optic neuropathy might be a consequence of the prone position. It is understood that in the early stages of the condition, changes to the peripapillary retinal nerve fiber layer (p-RNFL) and the macular ganglion cell layer and inner plexiform layer (m-GCIPL) can be detected quickly and accurately through the use of optical coherence tomography (OCT) [7]. During the COVID-19 pandemic, the use of the prone position grew with the rise in the number of severe COVID-19 cases, and we would therefore anticipate a corresponding increase in the number of ischemic optic neuropathy patients. Moreover, several published studies have detailed eye problems associated with COVID-19. Although alterations to the p-RNFL and m-GCIPL have been reported in COVID-19 patients, the observations of such cases lack consistency, and the underlying mechanisms and causes remain unknown [8–11].

This prospective, longitudinal, case-control study compared the thicknesses of the p-RNFL and m-GCIPL of COVID-19 patients placed in the prone position and those who were not in the prone position. Furthermore, any changes observed during the 6-month recovery phase were compared with a healthy control group.

## Methods

The study was carried out at the Neuro-Ophthalmology Clinic of the Department of Ophthalmology, Siriraj Hospital, Bangkok, from September 2021 to June 2022, with the prior approval of the Siriraj Institutional Review Board. All patients gave their written informed consent to participate, and the study fully complied with the Declaration of Helsinki. The participants were separated into 3 groups: a "Control Group" and 2 study groups: a "Prone Group" and a "Non-Prone Group." The Prone Group comprised COVID-19 patients in the prone position, whereas the Non-Prone Group consisted of COVID-19 patients who were not in that position. This pilot study set out to enroll 25 people in each of the 3 groups, as this was our first time using OCT to examine COVID-19 patients in the prone position.

The Prone Group and Non-Prone Group had the following inclusion criteria: participants aged 18–70 years old, confirmed COVID-19 diagnosis by a positive reverse transcription-polymerase chain reaction (RT-PCR) test via nasopharyngeal swab at Siriraj Hospital between September 1, 2021, and January 1, 2022, and recovery from COVID-19 (2–4 weeks following a negative RT-PCR test).

The criteria for exclusion were as follows:i.) ophthalmic conditions impacting the neuroretina, which encompass peripapillary retinal nerve fiber layer (p-RNFL) and macular ganglion cell-inner plexiform layer (m-GCIPL) thicknesses. These conditions include glaucoma, macular disease, uveitis, retinal vascular disorders, optic nerve disease, myopia with a refractive error surpassing 4 diopters, previous intraocular surgery, cataracts, or intraocular pressure exceeding 21 mmHg. ii.) neurological or systemic conditions affecting p-RNFL and m-GCIPL thicknesses, such as demyelinating disease, Alzheimer's disease,and Parkinson's disease. iii.) clinical status instabilities that presented difficulties during the investigation or unable to cooperate or remain seated for long periods. vi.) participants who became reinfected with

COVID-19, developed any of the above-excluded health conditions, required eye surgery, or no longer wished to participate.

Participants in the Control Group were drawn from the hospital database for 2018, during which period there were no cases of COVID-19. They were matched by sex and age with the participants in the 2 other groups. The Control Group participants had no ocular disease or systemic conditions and reported to the ophthalmology clinic to undergo annual routine ocular evaluations.

Each participant in the Prone Group and the Non-Prone Group completed an ophthalmic examination, encompassing best-corrected visual acuity (BCVA) testing measured in logMAR using the Early Treatment Diabetic Retinopathy Study (ETDRS) chart at 4 m. The IOP measurements were carried out with a Goldmann applanation tonometer, and any refractive errors were assessed via an automated refractometer. Color vision was tested with Ishihara pseudoisochromatic plates, the ocular surface and anterior segment were assessed with slit lamp biomicroscopy, and a dilated stereoscopic fundus examination was performed. All examinations were conducted by 1 physician (A.E.).

Visual field testing of participants' non-dilated pupils was carried out using the Humphrey Field Analyzer (Carl Zeiss Meditec Inc, Dublin, CA, USA) with the Swedish interactive threshold algorithm standard 24–2 program. Under the standard protocols, the study initially corrected the individual refractive error and near task, while the visual field findings met the reliability criteria of a fixation loss $< 20\%$ and a false-positive/false-negative error $< 15\%$.

An experienced technician conducted OCT (Cirrus OCT: OCT-3, OCT 6.0 software; Carl Zeiss Meditec Inc, Dublin, CA, USA) to record the thicknesses of the p-RNFL and m-GCIPL. A circular scan using a diameter of 3.4 mm was carried out to assess p-RNFL thicknesses, focusing on the optic disc with segments denoted as the superior, nasal, inferior, and temporal quadrants. The m-GCIPL thicknesses were established based on an area of 14.13 mm$^2$ centered upon the fovea. A ganglion cell analysis algorithm was used, and 6 different sectors were considered: superior, inferonasal, superonasal, inferior, inferotemporal, and superotemporal. OCT was determined to offer good quality $> 20$, and there was no significant difference in the OCT quality of the Prone Group and the Non-Prone Group ($P = 0.75$).

To determine macular thickness, SD-OCT (Heidelberg Engineering, Heidelberg, Germany) was used, while the macular area was examined by considering 3 concentric rings based upon the fovea: central ring (1 mm), inner ring (1–3 mm), and outer ring (3–6 mm). The analysis required 8 sectors to be separated from the rings: inner superior, outer superior, inner inferior, outer inferior, inner temporal, outer temporal, inner nasal, and outer nasal.

In addition, a fundus camera was used for fundus photography, while OCT was used to assess fundus autofluorescence.

In the Non-Prone Group, the ocular examinations were repeated within 30 days of each patient producing a negative RT-PCR result, with repeat examinations conducted at 3 and 6 months after. Regarding the Prone Group, as patients in an intensive care unit typically have a more extended recovery period, the analysis for the Prone Group was conducted first at 3 months and repeated at 6 months. A single physician gathered the data in the form of case records.

## Statistical analysis

Data for the initial examinations and the 3- and 6-month follow-ups of the Prone Group and the Non-Prone Group were compared. The data were also compared with the corresponding values of the Control Group at the same time points.

Each participant provided data from the right eye, and the statistics used to assess the quantitative variables were means, standard deviations, medians, ranges, and percentages. The

Kolmogorov–Smirnov test was employed to determine whether the data were normally distributed. For the case-control study, normally distributed variables were assessed via independent t-tests and paired t-tests to evaluate the results for the prospective cohort study over the 6-month study period. For data that did not follow a normal distribution, the Mann–Whitney U test was used in the case-control study, whereas the prospective cohort study employed the Wilcoxon signed-rank test. Spearman's rank correlation was used to evaluate the relationships between pairs of factors.

In all analyses, a *P* value < 0.05 indicated statistical significance. The analyses were carried out with PASW Statistics for Windows, version 18.0 (SPSS Inc, Chicago, IL, USA).

## Results

The Prone Group, Non-Prone Group, and Control Group each initially enrolled 25 participants. However, the exclusion criteria eliminated 3 patients from the Prone Group and 2 from the Non-Prone Group as they failed to attend the follow-up sessions. A further 3 patients were lost from the Prone Group as they could not sit still to enable satisfactory scan qualities to be obtained. Another 4 participants in the Prone Group died due to long COVID-19 sequelae. Consequently, the statistical analysis included data from 61 participants. The Non-Prone Group had 23 eyes (14 females; 9 males), the Prone Group had 15 eyes (10 females; 5 males), and the Control Group had 23 eyes (14 females; 9 males; *P* = 0.717).

The Control Group members were selected based on age and sex to match the Prone and Non-Prone Groups. The Non-Prone Group had a mean age of 51.61 ± 13.81 years (range, 31 to 76), while the Prone Group had a mean age of 57.33 ± 8.97 years (range, 42 to 71; P = 0.130). The median refraction was -0.50 diopters [D] (range, -3.75 to +4.00) for the Non-Prone Group and +1.00 D (range, 0.00 to +4.00) for the Prone Group. Underlying diseases were significantly more frequent in the Prone Group (hypertension, *P* = 0.030; dyslipidemia, *P* = 0.039).

The most frequently observed COVID symptoms were anosmia (56.5% for the Non-Prone Group; 86.7% for the Prone Group), with the 2 groups presenting no significant difference (*P* = 0.077). Symptoms of ageusia were significantly more common in the Prone Group (60%) than in the Non-Prone Group (4.3%; *P* < 0.001). Although headaches were also more common in the Prone Group (80.0%) than in the Non-Prone Group (30.4%), the difference was not significant (*P* = 0.007). All patients in the Prone Group exhibited severe clinical severity (100%). In contrast, most patients in the Non-Prone Group (21/23; 91%) were classified as having mild clinical severity, with none having severe clinical severity (**Table 1**).

The prospective longitudinal study revealed a mean BCVA of 20/20–20/25 and normal IOP in all groups. There were no significant differences in the BCVA and IOP of the Non-Prone Group and the Prone Group during the follow-up period. The color vision tests were normal for both study groups at each visit. No abnormal anterior segment or vitreous findings were found in any patient. No abnormal findings were observed using fundus autofluorescence (**Table 2**).

In the Prone Group, the initial fundus examinations revealed 33.33% abnormality (5/15 patients), while in the Non-Prone Group, the proportion was 4.35% (1/23 patients). The first Prone Group case had multiple cotton-wool spots (CWSs) that were 50% resolved at 6 months. The second Prone Group case had multiple CWSs with blot retinal hemorrhage; both conditions fully cleared within 6 months. The third case had a splinter hemorrhage with mild tortious retinal veins that were no longer visible at 6 months. The fourth Prone Group case exhibited a small retinal exudate near the optic disc with moderate tortuous retinal vessels. This problem was fully resolved within 6 months. The fifth case showed a ghost vessel of a

**Table 1. Demographic data of patients.**

|  | None-prone group N = 23 | Prone group N = 15 | *p-value* |
|---|---|---|---|
| Age, year Mean (SD) | 51.61 (13.81) | 57.33 (8.97) | 0.130 |
| Sex, male, No (%) | 9 (39.1) | 5 (33.3) | 0.717 |
| Refraction, Median(rang) | -0.50 (-3.75,+4.00) | +1.00 (0.00,+4.00) | - |
| **Medical history** |  |  |  |
| Hypertension, No (%) | 3 (15.8) | 7 (46.7) | **0.030** |
| Dyslipidemia, No (%) | 2 (10) | 6 (42.9) | **0.039** |
| **Severity of COVID-19** |  |  |  |
| Mild, No (%) | 21 (91) | 0 | - |
| Moderate, No (%) | 2 (9) | 0 | - |
| Severe, No (%) | 0 | 15 (100) | - |
| **Symptoms** |  |  |  |
| Anosmia, No (%) | 13 (56.5) | 13 (86.7) | 0.077 |
| Ageusia, No (%) | 1 (4.3) | 9 (60) | < **0.001** |
| Headache, No (%) | 7 (30.4) | 12 (80) | 0.007 |

SD: Standard deviation, Significant differences are shown in bold.

small branch in the papillomacular bundle area that prevailed for at least 6 months. Regarding the single case in the Non-Prone Group, the patient had a small blot retinal hemorrhage close to the optic disc that lasted just 3 months (**Table 2**).

The prospective longitudinal study included an OCT analysis to draw comparisons within groups over the 6-month study period. In the case of the p-RNFL thicknesses, no significant

**Table 2. Comparison of BCVA, IOP, color vision, fundus examination and fundus autofluorescence in Covid group, Prone group, and Control group.**

|  |  | BCVA LogMAR Median(rang) | IOP (mmHg) Mean (SD) | Color vision Normal % | Fundus examination number of Abnormal case (%) | Fundus autofluorescence Normal % |
|---|---|---|---|---|---|---|
| **Non-prone group** n = 23 | 1 mo. | 0.0 (0.0–0.5) | 13.7 (2.80) | 100% | 1(4.35) | 100% |
|  | 3 mo. | 0.1 (0.0–0.5) | 13.8 (3.30) | 100% | 0 | 100% |
|  | 6 mo. | 0.0 (0.0–0.4) | 13.7 (3.11) | 100% | 0 | 100% |
|  | p | 0.314 | 0.786 | - | - | - |
| **Prone group** n = 15 | 3 mo. | 0.1 (0.0–0.6) | 15.73 (2.19) | 100% | 5 (33.33) | 100% |
|  | 6 mo. | 0.1 (0.0–0.6) | 15.47 (2.17) | 100% | 1 (6.67) | 100% |
|  | p | 0.257 | 0.334 | - | - | - |
| **Control** n = 23 |  | 0.1 (0.0–0.2) | 13.96 (2.46) | 100% | 0 | 100% |
| P value between group | 3 mo. | 0.723 | 0.052 | - | - | - |
|  | 6 mo. | 0.658 | 0.057 | - | - | - |

BCVA: Best corrected visual acuity, IOP: Intraocular pressure, SD: Standard deviation.

Significant differences are shown in bold.

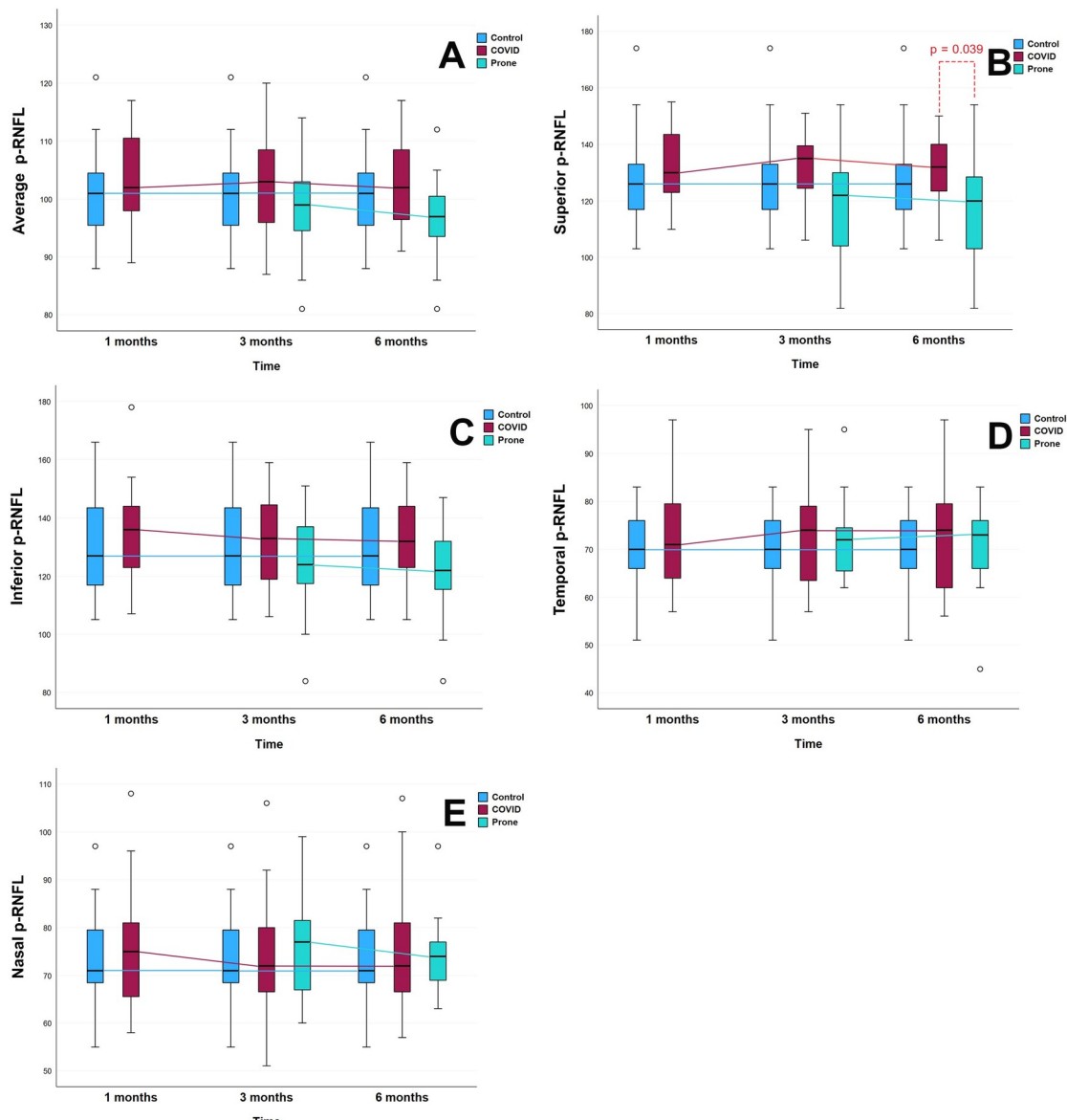

**Fig 1. Demonstrate the comparison of p-RNFL thickness in the None-prone group (COVID), prone group, and control group at 3 and 6 months.** (A) is an average of p-RNFL, (B) is a superior segment of p-RNFL, (C) is an inferior segment of p-RNFL, (D) is a temporal segment of p-RNFL, and (E) is a nasal segment of p-RNFL.

differences were reported within the Prone Group and the Non-Prone Group at different times (**Fig 1**). However, the m-GCIPL analysis revealed significant differences in the nasal-inferior sector of the Non-Prone Group between months 1 and 3 (mean difference, 0.74; $P = 0.009$; **S1 Table**). In contrast, the Prone Group presented no significant differences between the 3-month and 6-month values for m-GCIPL thickness (**Fig 2**). The macular thickness analysis for the Non-Prone Group found no significant differences between months 1 and 3 (**Table 3**).

An OCT analysis was performed as part of the case-control study to compare the different groups (**S2 and S3 Tables**). The analysis showed that at 1 month, The m-GCIPL analysis revealed that the inferior sector of the Non-Prone Group ($80.57 \pm 4.60$ µm) was significantly

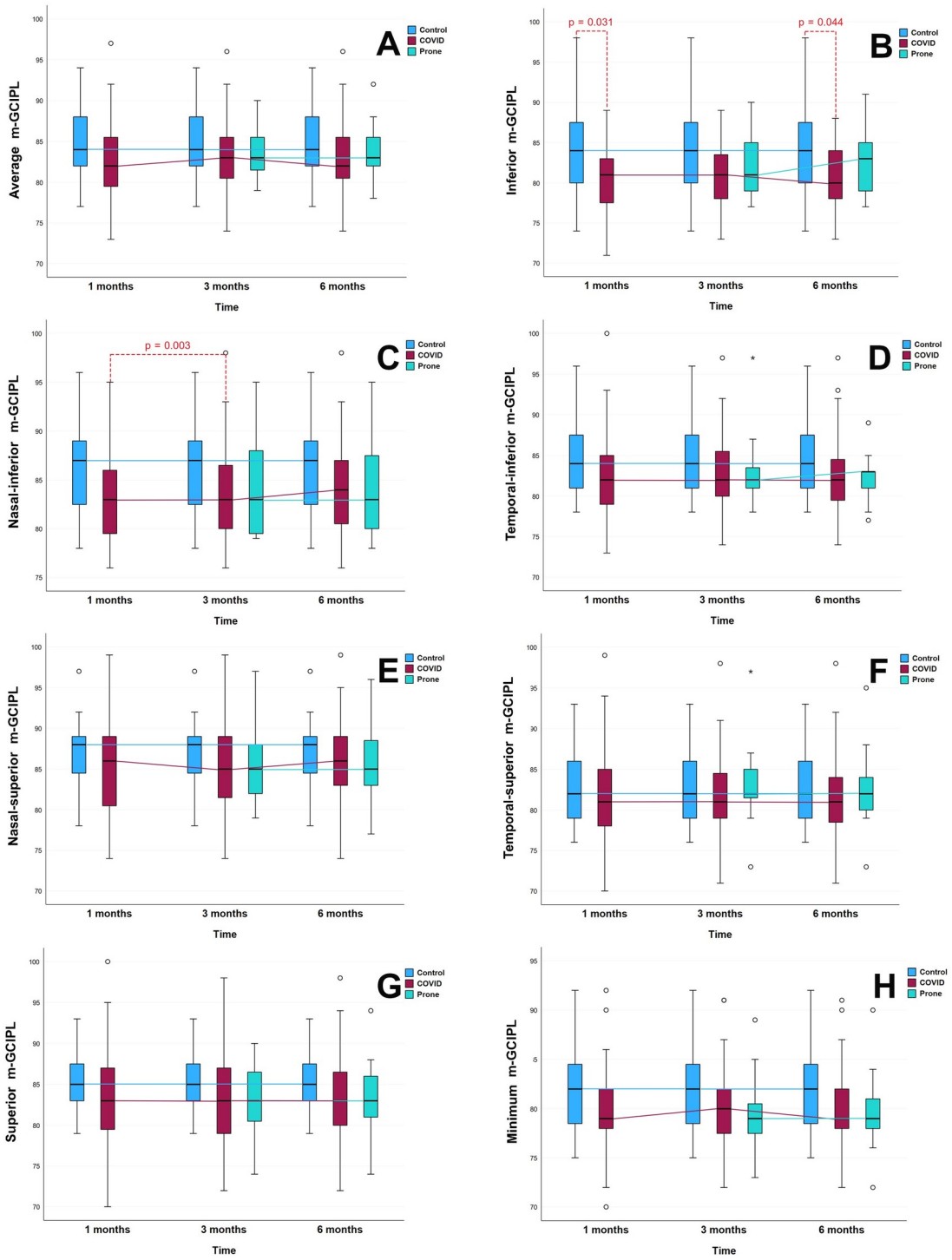

**Fig 2. Demonstrate the comparison of m-GCIPL thickness in the None-prone group (COVID), prone group, and control group at 3 and 6 months.** (A) is an average of m-GCIPL, (B) is an inferior segment of m-GCIPL, (C) is a nasal-inferior segment of m-GCIPL, (D) is a temporal-inferior segment of m-GCIPL, (E) is a nasal-superior segment of m-GCIPL, (F) is a temporal-superior segment of m-GCIPL, (G) is a superior segment of m-GCIPL, and (H) is a minimum of m-GCIPL.

**Table 3. (6): Comparison of macular thickness between None-prone group and Prone group.**

| Macular thickness | Non-prone group (n = 23) | | | Prone group (n = 15) | p-value between group at 3 mo. | Mean differences | | 95% CI | |
|---|---|---|---|---|---|---|---|---|---|
| | 1 mo. Mean (SD) | 3 mo. Mean (SD) | p-value in group | 3 mo. Mean (SD) | | Between 1–3 mo in COVID-19 group | Between None-prone–Prone group at 3 mo. | Between 1–3 mo in None-prone group | Between None-prone–Prone group at 3 mo. |
| C (μm) | 266.83 (23.27) | 267.87 (22.02) | 0.277 | 260.00 (19.27) | 0.266 | -1.04 | 7.87 | -2.98 to 0.90 | -6.26 to 22.00 |
| SI (μm) | 336.22 (18.71) | 337.43 (20.12) | 0.058 | 333.53 (19.80) | 0.560 | -1.22 | 3.90 | -2.48 to 0.05 | -9.56 to 17.36 |
| NI (μm) | 339.91 (17.43) | 340.13 (18.60) | 0.778 | 332.40 (12.89) | 0.169 | -0.22 | 7.73 | -1.80 to 1.36 | -3.45 to 18.91 |
| II (μm) | 335.48 (17.91) | 335.26 (16.09) | 0.812 | 326.33 (10.36) | **0.045** | 0.22 | **8.93** | -1.65 to -2.09 | **0.23 to 17.63** |
| TI (μm) | 323.78 (17.72) | 325.57 (17.55) | 0.144 | 317.60 (15.48) | 0.161 | -1.78 | 7.97 | -4.23 to 0.66 | -3.33 to19.26 |
| SO (μm) | 301.13 (15.25) | 301.26 (15.99) | 0.803 | 293.33 (10.26) | 0.098 | -0.13 | 7.93 | -1.20 to 0.94 | -1.52 to 17.38 |
| NO (μm) | 316.00 (13.18) | 314.39 (17.35) | 0.338 | 309.27 (12.21) | 0.328 | 1.61 | 5.13 | -1.80 to 5.02 | -5.35 to 15.60 |
| IO (μm) | 287.30 (12.19) | 287.26 (13.25) | 0.936 | 284.87 (10.81) | 0.563 | 0.04 | 2.39 | -1.07 to 1.15 | -5.93 to 10.71 |
| TO (μm) | 283.7 (15.50) | 285.61 (14.99) | 0.287 | 283.07 (7.92) | 0.501 | -1.91 | 2.54 | -5.55 to 1.72 | -5.04 to 10.13 |

C: Central, SI: Superior-inner, NI: Nasal-inner, II: Inferior-inner, TI: Temporal-inner, SO: Superior-outer, NO: Nasal- outer, IO: Inferior- outer, TO: Temporal- outer, SD: Standard deviation, CI: Confidence interval, Significant differences are shown in bold.

thinner than that of the Control Group (83.87 ± 5.43 μm), with a mean difference of 3.30 μm ($P$ = 0.031) but there was no significant difference between the p-RNFL. The values analysis of the p-RNFL and m-GCIPL values at 3 months found no significant differences among the 3 groups. However, the analysis of macular thickness at 3 months revealed significantly lesser thickness for the inferior-inner sector of the Prone Group (326.33 ± 10.36 μm) than the Non-Prone Group (335.26 ± 16.09 μm), with a mean difference of 8.93 μm ($P$ = 0.045; **Table 3**). Moreover, the p-RNFL analysis showed significantly lesser thickness at 6 months for the superior sector of the Prone Group (118.87 ± 18.21 μm) than the Non-Prone Group (131.61 ± 12.08 μm), with a mean difference of 12.74 μm ($P$ = 0.039). In addition, the significant difference of inferior sector of m-GCIPL between the Non-Prone Group and control group was found at 6 month by the Non-Prone Group (80.48 ± 3.96 μm) was significantly thinner than that of the Control Group (83.87 ± 5.43 μm), with a mean difference of 3.39 μm ($P$ = 0.044).

The limitation were insufficient data to perform the m-GCIPL, p-RNFL, and macular thickness analysis at 1 months of the Prone Group and at 6 months of the boths.

In the Prone Group, the median time of prone positioning was 49.1 hours (interquartile range, 16, 128). A subgroup analysis revealed no correlation between the number of days of prone positioning and changes in the p-RNFL or m-GCIPL thicknesses.

When the visual field parameters (visual field index, mean deviation, and pattern standard deviation) of the Prone and Non-Prone Groups at different time points were compared, no significant differences were observed **S4 Table**.

## Discussion

Alterations in p-RNFL thickness can be detected early in retina and optic nerve diseases. Only 1 prospective study [12] exhibited statistically significant thinning of p-RNFL thicknesses in the inferior (P = 0.009) and nasal (P = 0.003) quadrants between before and after surgery in patients who underwent spinal surgery while placed in the prone position.

This research is the first prospective long-term study to assess p-RNFL thickness changes in COVID-19 patients in prone positions. No significant differences in p-RNFL could be observed at 3 months in either the Prone Group or the Non-Prone Group. Furthermore, no difference was reported for the m-GCIPL, which typically shows faster changes than the p-RNFL.

The macular thickness evaluation showed that the inferior-inner sector was significantly thinner in the Prone Group than in the Non-Prone Group.

At the 6-month follow-up, the superior sector of the p-RNFL was significantly thinner in the Prone Group than in the Non-Prone Group. In contrast, the analysis of m-GCIPL thicknesses revealed no significant differences. Since the data were incomplete, it was impossible to carry out the macular thickness analysis at 6 months. Our study was further limited by its sample size; a larger population would have been preferable.

However, the p-RNFL thickness of the Prone Group tends to be thinner than that of the Non-Prone Group, but the m-GCIPL thicknesses are not different **(Figs 1 and 2).**

The prone position affects the alteration of neuroretina in various ways. First, the position leads to a greater IOP and a lower mean arterial pressure. Consequently, the blood flow to the optic nerve falls since the flow is a function of the difference between the IOP and the mean arterial pressure [13,14]. Furthermore, extended use of the prone position can result in periorbital edema, which elevates the orbital venous pressure and subsequently increases the IOP. Support by our study found that the mean IOP of the Prone Group was 15.73 ± 2.19 mmHg at 3 months, significantly exceeding the value of the Non-Prone Group (13.80 ± 3.30 mmHg; P = 0.052). Similarly, the mean IOP value at 6 months for the Prone Group (15.47 ± 2.17 mmHg) was significantly greater than that for the Non-Prone Group (13.70 ± 3.11 mmHg; P = 0.057). The sustained elevation of intraocular pressure (IOP) observed in the Prone group is hypothesized to stem from prolonged prone positioning. This positioning may induce venous stasis and permanent venous dilatation, subsequently augmenting episcleral venous pressure and resulting in sustained elevated IOP levels **Table 2.**

One unusual and rare complication after prone position in spine surgery is perioperative ischemic optic neuropathy (ION), which has an incidence of approximately 0.2% [15–19]. Additionally, a major review [5] showed that a mean prone positioning duration of 497 ± 180 minutes was sufficient to bring about ION after spinal surgery. Six out of 58 patients (10.3%) developed ION despite not being placed in the prone position. However, a case-control analysis concluded that prone positioning could not be considered an independent risk factor for ION (P = 0.77).

Once the COVID-19 pandemic struck, patients on ventilators in intensive care units were placed in the prone position to address the problem of acute respiratory failure. It was suggested that this should be done for 12–16 hours daily [20,21]. Patients placed regularly in the prone position for long periods, ranging from 1–5 days on a case-by-case basis, as well as COVID-19 patients who are severely ill.

Furthermore, vasopressors can lead to vasoconstriction and vasospasms of the ophthalmic and ciliary vessels, causing disruptions to the blood flow to the optic nerve and requiring hemodialysis or fluid replacement. These conditions can directly lead to ION [22,23]. It can be inferred that their causes are responsible for the heightened incidence of ION rather than

perioperative ION. Today, however, 4 years after the COVID-19 pandemic began, ION has not been reported in patients who had been positioned in the prone position. Our research also found no evidence of loss of visual function following prone positioning. However, there may be some underestimation since it can be challenging to diagnose ION in the context of critical care, with this difficulty exacerbated when patients are unable to explain their vision problems verbally. One sign that can be detected is a relative afferent pupillary defect; however, it can still be easily overlooked.

The viral infection and immune reaction involved in COVID-19 affect multiple organs. The pathogenesis includes tissue ischemia and microvascular dysfunction, which result from endotheliitis. The angiotensin-converting enzyme-related carboxypeptidase (ACE2) receptor serves as the site of entry for the virus. It is then expressed in Muller cells, the pericytes of endothelial cells in the retina and choroid, and the retinal pigment epithelium [24]. Ocular complications are found in 2% to 32% of patients with COVID-19 [25]. Problems commonly affect the anterior segment, typically involving conjunctivitis or anterior uveitis. However, some reports have mentioned more severe conditions, such as retinitis, choroiditis, optic neuritis, and retinal vasculitis [26–29].

It is also possible that the ACE2 receptor might be involved in cases of non-ischemic damage to the retina [30]. The virus recruits immune cells in vessel walls, which can lead to edematous endothelial cells [31]. Furthermore, the virus can induce an immune response that can cause endothelial dysfunction and apoptosis, leading to microthrombotic events. The steady flow of blood to the retina is resistant to autonomic innervation and hormonal mediators, making the retina ideal for examining the local microcirculation.

The almost of the previous studies reported that in the initial 4 weeks after COVID-19 infection, the neuroretinal changes in COVID-19 patients showed significantly thicker p-RNFL than in controls[7,8,10,32] same as our finding. This difference can be explained as a consequence of the inflammatory process resulting from the effects of the virus or the ischemic activity of the peripapillary capillary plexus. This effect was observed by Savastano et al. [10].

Our research showed that the average and segmental p-RNFL thicknesses tended to decrease over time in the Prone and the Non-Prone Groups but remained thicker than those in the controls at all time points that was supported by previous report [33,34]. (Fig 1) One exception to this was that the superior and inferior segments for the Prone Group were not as thick as those for the Control Group, although the difference was not significant. (Fig 1A and 1B) At 6 months, superior segment thinning in the Prone Group had decreased significantly compared with that in the Non-Prone Group. In addition, 4 case series (3 from the Non-Prone Group and 1 from the Prone Group) underwent OCT baseline checks at least 6 months before the onset of COVID-19. Their data showed that all their p-RNFL segments were thinner than the post-COVID-19 measurements, although no assessment was made of the statistical significance.

It can be supposed that the inflammation of the retina associated with COVID-19 influences p-RNFL, leading to the initial edema of p-RNFL, whereby OCT identified an increase in thickness, which then gradually thinned as time passed. Edematous p-RNFL persisted in COVID-19 patients for at least 6 months, exceeding the duration of alternative etiologies, possibly resulting from long-term occult inflammation rather than a full recovery of clinical status. Follow-up visits over the longer term would be necessary to identify any additional retinal changes. According to the literature, the prone position is linked to an increase in IOP; in our study, a loss of p-RNFL resulted, leading to the observation of reduced thickness via OCT examination. In particular, this was important for the superior and inferior segments since they are the most vulnerable components of the structure.

The reported changes in m-GCIPL thickness have not been consistent, with 2 studies finding a reduced thickness [32,35], whereas another study showed the opposite [36].

In our research, there was no tendency for m-GCILP thickness to change over time, whether by mean or by individual sector, in either the Prone or the Non-Prone Group (Fig 2). While a statistically significant difference was observed in the values of the inferior sector of the Non-Prone Group between 1 and 3 months after infection, with a mean difference of 0.74 μm (P = 0.008), the result could not be considered clinically significant Fig 2C. Compared with the control subjects, the m-GCILP thicknesses of the Prone and Non-Prone Groups showed trends toward becoming thinner, although without significance. Notably, at 6 months, the Non-Prone Group had a significantly thinner inferior segment than the Control Group, with a mean difference of 3.391 μm (P = 0.044) Fig 2B. However, the 4-case series in this study for which baseline OCT data were available showed no patterns of change in m-GCILP.

The retinal features in COVID-19 patients most commonly described are CWSs, venous tortuosity, and microhemorrhages. These features match this study's results, and all are indicative of acute vascular events and retinal ischemia. It has been hypothesized that the underlying reasons include occlusive vasculopathy (microthrombolic events), hypercoagulopathy, ACE2 downregulation by the COVID-19 virus, or immune-complex deposition within the vessel walls [37]. The current investigation suggests that the Prone Group patients who had experienced more severe COVID-19 and had undergone intensive care showed more abnormalities in the fundus than patients from the Non-Prone Group. We argue that disease severity and certain intensive care treatment protocols might have significantly shaped our study's findings. Numerous studies have observed the CWSs presence 7.4–22% of both active COVID-19 patients and those in the early recovery stage that supported our finding [38,39]. Additionally, Invernizzi et al. [40] found CWSs in 7.4% of the patients in a study comprising 54 COVID-19 patients. Retinal hemorrhages were observed in 9.25% of the patients, while dilated retinal veins were more common (27.7%). Finally, Lani-Louzada et al. [41] studied 25 COVID-19 patients and found retinal hemorrhages in 12%. However, research by Pareddy et al. [42] conducted 6 days after the onset of COVID-19 in a large cohort found nothing except one streak hemorrhage in a single patient. Moreover, retinal hemorrhages, particularly splinter hemorrhages, and tortuous retinal veins can be detected in cases of venous stasis resulting from prone positioning. It is important to note that these ocular manifestations are not attributed to COVID-19 infection.

Most dyschromatopsia cases associated with COVID-19 can be linked to optic neuritis evidence. One 53-year-old male presented with isolated dyschromatopsia in the left eye, characterized by acute vision loss and scotoma [43]. Numerous retinal hemorrhages were found on his fundus, while OCT revealed acute macular neuroretinopathy and paracentral acute middle maculopathy. However, in the present study, no cases of dyschromatopsia were observed among the COVID-19 patients.

This study marks the first occasion in which visual field parameters (visual field index, mean deviation, and pattern standard deviation) have been examined in the context of COVID-19. Our findings revealed no significant differences. We can therefore infer from the OCT analysis that any structural changes in COVID-19 patients will be identified more rapidly than functional changes (visual field), in contrast to glaucoma.

The limitation in this study was a small sample size. Recruitment of the Prone Group proved difficult because all patients were in the intensive care unit with critical illnesses and had high morbidity and mortality. Moreover, the research was conducted during the COVID-19 pandemic, which resulted in many potential subjects declining to participate. Furthermore, the severity of the disease was markedly higher in the Prone Group, primarily due to the natural characteristic of individuals who had to assume the prone position, potentially affecting

neuroretinal changes. Although statistical analysis did not account for this difference, this study represents the first pilot investigation into this issue. Hence, a novel research endeavor with a more concise methodology is warranted. Additionally, we suggest employing OCT angiography for further studies, as it is more suitable for evaluating ischemic changes.

In conclusion, the current study represents the first longitudinal study on the use of the prone position for COVID-19 patients. Our findings indicate that COVID-19 and the use of the prone position can lead to neuroretinal changes. The prone position may lead to an early loss of p-RNFL thickness due to rising IOP, which is independent of the timing of prone positioning. No change was observed in the m-GCIPL. Since there is no increase in the morbidity burden, we argue that using the prone position is vital in lowering mortality rates among patients with severe COVID-19. One consequence of COVID-19 is the onset of edematous p-RNFL during the active phase of the disease, followed by subsequent resolution during the recovery phase. In comparison with the Control Group, no thinning by atrophy was observed during the 6-month study period in the Non-Prone Group, but it was evident in the Prone Group at 3 and 6 months. The thinning of the m-GCIPL by atrophy was observed in both the COVID-19 groups when compared with the control group. However, conducting trials involving more participants and extended follow-up periods may be necessary to determine the exact nature of retinal changes over time. However, it may be necessary to conduct trials involving more participants and extended follow-up periods to determine the exact nature of retinal changes over time.

## Supporting information

**S1 Table. Comparison of m-GCIPL in None-prone group in different times.** mGCIPL: Macular ganglion cell+inner plexiform layer, Av: Average, Min: Minimum, NS: Nasal-superior, NI: Nasal-inferior, S: Superior, I: Inferior, TS: Temporal-superior, TI: Temporal-inferior, SD: Standard deviation, CI: Confidence interval, Significant differences are shown in bold.
(DOCX)

**S2 Table. Comparison of p-RNFL thickness in None-prone group, Prone group, and Control group at 3 and 6 month.** None-prone: None-prone group, Prone: Prone group, Control: Control group, pRNFL: Peripapillary RNFL, Av: Average, S: Superior, N: Nasal, I: Inferior, T: Temporal, SD: Standard deviation, CI: Confidence interval, Significant differences are shown in bold.
(DOCX)

**S3 Table. Comparison of m-GCIPL in None-prone group, Prone group, and Control group at 3 and 6 month.** None-prone: None-prone group, Prone: Prone group, Control: Control group, mGCIPL: Macular ganglion cell+inner plexiform layer, Av: Average, Min: Minimum, NS: Nasal-superior, NI: Nasal-inferior, S: Superior, I: Inferior, TS: Temporal-superior, TI: Temporal-inferior, SD: Standard deviation, CI: Confidence interval, Significant differences are shown in bold.
(DOCX)

**S4 Table. Comparison of visual field parameters in None-prone group and Prone group at 1, 3 and 6 month.** None-prone: None-prone group, Prone: Prone group, VFI: Visual field index, MD: Mean deviation, PSD: Pattern standard deviation, SD: Standard deviation, Significant differences are shown in bold.
(DOCX)

## Acknowledgments

The authors offer sincere gratitude to Associate Professor Nithipatana Chierakul and the Department of Critical Care Medicine in the Faculty of Medicine, Siriraj Hospital, for recruiting patients for this study. Moreover, thanks are due to Pawipon Nisarat, MD, for the original idea of this study. The authors are also indebted to David Park for the English-language editing of this paper. Finally, the authors thank the study participants, who made the research possible.

## Author Contributions

**Conceptualization:** Akarawit Eiamsamarng.

**Data curation:** Akarawit Eiamsamarng.

**Funding acquisition:** Niphon Chirapapaisan.

**Investigation:** Niphon Chirapapaisan.

**Methodology:** Niphon Chirapapaisan, Natthapon Rattanathamsakul, Ranistha Ratanarat.

**Project administration:** Akarawit Eiamsamarng.

**Resources:** Ranistha Ratanarat.

**Software:** Natthapon Rattanathamsakul.

**Supervision:** Niphon Chirapapaisan, Wanicha Chuenkongkaew, Ranistha Ratanarat.

**Validation:** Niphon Chirapapaisan, Wanicha Chuenkongkaew, Natthapon Rattanathamsakul.

**Visualization:** Natthapon Rattanathamsakul.

**Writing – original draft:** Akarawit Eiamsamarng.

**Writing – review & editing:** Niphon Chirapapaisan, Wanicha Chuenkongkaew.

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
