## [Decision Letter · Decision Letter 0]

15 Feb 2024

PONE-D-24-02182The prone position in COVID-19 impacts the thickness of peripapillary retinal nerve fiber layers and macular ganglion cell layers: a prospective longitudinal studyPLOS ONE

Dear Dr. Eiamsamang,

Thank you for submitting your manuscript to PLOS ONE. After careful consideration, we feel that it has merit but does not fully meet PLOS ONE’s publication criteria as it currently stands. Therefore, we invite you to submit a revised version of the manuscript that addresses the points raised during the review process.

**ACADEMIC EDITOR:**

Since the main hypothesis of the study is ischemic optic neuropathy, OCT angiography could have been the ideal to assess these cases .( Reviewer 1). Ischemic changes , at least in the early phases, can lead to axonal swelling and thence increase pRNFL thickness and mGCC thickness . 

Visual Field test was mentioned in methodology  in full details  , however no mention about the results ( Reviewer 1 ). Severity of the disease in prone Group is higher than the non-prone Group . Statistical analysis should compensate for this 

Cases with splinter hemorrhage  and tortuous retinal veins : implying venous stasis , or impending vein occlusion. This can affect pRNFL and m GCC thickness 

IOP rise related to prone position, needing more explanation in discussion why significant difference persists 6 months after recovery   

We look forward to receiving your revised manuscript.

Kind regards,

Karim Adly Raafat

Academic Editor

PLOS ONE

Journal Requirements:

"Siriraj Research Fund, Faculty of Medicine, Siriraj Hospital, Mahidol University (grant number IO–R016531014.)"

Reviewers' comments:

Reviewer's Responses to Questions

**Comments to the Author**

1. Is the manuscript technically sound, and do the data support the conclusions?

Reviewer #1: Yes

Reviewer #2: Yes

2. Has the statistical analysis been performed appropriately and rigorously? 

Reviewer #1: Yes

Reviewer #2: Yes

3. Have the authors made all data underlying the findings in their manuscript fully available?

Reviewer #1: Yes

Reviewer #2: Yes

4. Is the manuscript presented in an intelligible fashion and written in standard English?

Reviewer #1: Yes

Reviewer #2: No

5. Review Comments to the Author

Reviewer #1: The manuscript is well written and the interpreation of the findings sound. I just have 2 comments

1- The manuscript would have benefited from performing OCT angiography to study the ischemic changes if present. This could be added to the limitations

2- I cant find the results of the visual field tesing.

Reviewer #2: The research question is discussing a very novel point

The methodology and data analysis are presented in a very good manner

The discussion section requires revision and editing for data to be presented in a more organized and explanatory style

6. PLOS authors have the option to publish the peer review history of their article (what does this mean?). If published, this will include your full peer review and any attached files.

Reviewer #1: **Yes: **Ahmed Awadein

Reviewer #2: No

---

## [Author Response · Author response to Decision Letter 0]

28 Feb 2024

They were shown in Respond to Reviewers Word file.

---

## [Editor Report · Decision Letter 1]

4 Mar 2024

The prone position in COVID-19 impacts the thickness of peripapillary retinal nerve fiber layers and macular ganglion cell layers

PONE-D-24-02182R1

Dear Dr. Eiamsamang,

We’re pleased to inform you that your manuscript has been judged scientifically suitable for publication and will be formally accepted for publication once it meets all outstanding technical requirements.

Kind regards,

Karim Adly Raafat, M.D.

Academic Editor

PLOS ONE